Distributional dynamics of a vulnerable species in response to past and future climate change: a window for conservation prospects

Bai Yunjun 1
Wei Xueping 2
Li Xiaoqiang lixiaoqiang@ivpp.ac.cn 1
1 Key Laboratory of Vertebrate Evolution and Human Origin of Chinese Academy of Sciences, Institute of Vertebrate Paleontology and Paleoanthropology, Chinese Academy of Sciences , Beijing
2 Key Laboratory of Bioactive Substances and Resources Utilization of Chinese Herbal Medicine, Engineering Research Center of Tradition Chinese Medicine Resource, Ministry of Education, Institute of Medicinal Plant Development, Chinese Academy of Medical Sciences, Peking Union Medical College , Beijing
Mucina Ladislav
Electronic publication date: 2018 Jan 16
Publication date: 2018
Volume: 6
Electronic Location ID: e4287
Received 2017 Oct 11; Accepted 2018 Jan 2
Copyright: ©2018 Bai et al.
Copyright year: 2018
Copyright holder: Bai et al.
License: This is an open access article distributed under the terms of the Creative Commons Attribution License, which permits unrestricted use, distribution, reproduction and adaptation in any medium and for any purpose provided that it is properly attributed. For attribution, the original author(s), title, publication source (PeerJ) and either DOI or URL of the article must be cited.
License URL: https://creativecommons.org/licenses/by/4.0/

Keywords: Climate change, Pseudolarix amabilis, MaxEnt, Range dynamics, Species conservation

Funding: National Natural Science Foundation of China 41372175 41602188 National Basic Research Program of China 2015CB953803 This work was supported by the National Natural Science Foundation of China (Nos. 41372175, 41602188) and National Basic Research Program of China (No. 2015CB953803). The funders had no role in study design, data collection and analysis, decision to publish, or preparation of the manuscript.

==============================
Background

The ongoing change in climate is predicted to exert unprecedented effects on Earth’s biodiversity at all levels of organization. Biological conservation is important to prevent biodiversity loss, especially for species facing a high risk of extinction. Understanding the past responses of species to climate change is helpful for revealing response mechanisms, which will contribute to the development of effective conservation strategies in the future.

Methods

In this study, we modelled the distributional dynamics of a ‘Vulnerable’ species, Pseudolarix amabilis, in response to late Quaternary glacial-interglacial cycles and future 2080 climate change using an ecological niche model (MaxEnt). We also performed migration vector analysis to reveal the potential migration of the population over time.

Results

Historical modelling indicates that the range dynamics of P. amabilis is highly sensitive to climate change and that its long-distance dispersal ability and potential for evolutionary adaption are limited. Compared to the current climatically suitable areas for this species, future modelling showed significant migration northward towards future potential climatically suitable areas.

Discussion

In combination with the predicted future distribution, the mechanism revealed by the historical response suggests that this species will not be able to fully occupy the future expanded areas of suitable climate or adapt to the unsuitable climate across the future contraction regions. As a result, we suggest assisted migration as an effective supplementary means of conserving this vulnerable species in the face of the unprecedentedly rapid climate change of the 21st century. As a study case, this work highlights the significance of introducing historical perspectives while researching species conservation, especially for currently vulnerable or endangered taxa that once had a wider distribution in geological time.

Introduction

Of the four billion species that have evolved on Earth over the past 3.5 billion years, 99% are considered to have disappeared (Novacek, 2001), most notably in the ‘Big Five’ mass extinctions (Raup & Sepkoski, 1982; Jablonski & Chaloner, 1994; Bambach, 2006). Considerable species losses over the past few centuries and millennia as a result of anthropogenic climate change have sounded the alarm about a possible sixth mass extinction (Barnosky et al., 2011). Effective conservation planning is necessary to avoid potential biodiversity loss, and an understanding of the distribution dynamics of organisms in response to climate change underlies the development of effective conservation strategies (Razgour et al., 2013).

Climate change is recognized as a key driver of species’ range dynamics, both locally and globally, over time (Huntley et al., 1995; Pearson & Dawson, 2003; Yates et al., 2010; Hamer et al., 2015). Quaternary climatic oscillations, particularly the most recent late Quaternary, characterized by markedly recurring glacial-interglacial cycles have played a crucial role in shaping the contemporary geographical distribution of plant species (Comes & Kadereit, 1998; Dynesius & Jansson, 2000; Hewitt, 2000; Sandel et al., 2011). The Last Interglacial (LIG, ∼120–140 ka) and Last Glacial Maximum (LGM, ∼21 ka) periods mark contrary extremes during the late Quaternary (Dawson, 1992), with the latter especially representing one of Earth’s most extreme periods of environmental variability (Clark et al., 2009). Indeed, the climate has warmed from the LGM to the present, and the temperature variations during this interval cover almost the entire temperature range of the Quaternary (Imbrie, McIntyre & Mix, 1989; Ruddiman, 2008). The dramatic climatic cooling during the LGM drove many species to glacial refugia (Nogués-Bravo et al., 2010), though populations began to recolonize during postglacial climate warming (Davis & Shaw, 2001; Normand et al., 2011).

As human activity intensifies, global temperatures are expected to rise by 1.1–6.4 °C during the 21st century (Intergovernmental Panel on Climate Change, 2007). Due to this unprecedentedly rapid rate of warming, climate change is predicted to be the greatest force in reshaping the geographical distribution of species in the 21st century (Leadley et al., 2010). Moreover, given the rapidity of climate change over the coming decades, whether populations can shift rapidly enough successfully tracking climate change is the central concern of many ecology studies (Davis & Shaw, 2001). Therefore, although many organisms have survived multiple climate cycles during their evolutionary histories (Meyers & Bull, 2002), some species are unable to disperse or adapt fast enough to track the rapidly changing climate, leading to increased extinction risk (Warren et al., 2001; Menéndez et al., 2006). In addition, landscape modifications resulting from the intensification of human activity may aggravate the negative effects of climate change by impeding species migration. Consequently, there is great concern about the challenges posed to extant species by the ongoing unprecedented change (Thuiller, 2007).

The response of species to climate change can be synthesized as evolutionary adaptation, dispersal or extinction (Parmesan, 2006; Aitken et al., 2008; Dawson et al., 2011), processes that are related to the velocity of the climate change and the species’ capacity to adapt and migrate (Jump & Penuelas, 2005; Sandel et al., 2011). Populations that are unable to keep up with climate change or to adapt to new climate conditions, especially those with narrow climatic tolerances (Thuiller, Lavorel & Araújo, 2005), face a very high risk of extinction (Hofreiter & Stewart, 2009; Dawson et al., 2011; Molinos et al., 2016). Endemic species are unique to a defined geographic range. These restricted-range species may be highly vulnerable to rapid climate change because of their narrow climatic tolerances (Malcolm et al., 2006; Ohlemüller et al., 2008), indicating that endemism and extinction risk are closely related (Petit, Hu & Dick, 2008). Therefore, surveying the response of endemic species to climate change is particularly important.

Pseudolarix amabilis is a representative of monotypic genus of the family Pinaceae; it is endemic to China, inhabiting a highly restricted area of the lower Yangtze River at an elevation range of 180–1,000 m (Yang & Christian, 2013). This deciduous tree’s branchlets are dimorphic: long branchlets (leading shoots) with helically borne leaves and short branchlets (brachioblasts) with fascicularly arranged leaves. The bract-scale complexes of the seed cones shed at maturity; two winged seeds, located at the base of the seed scales, mature in the 1st year (Fu, Li & Robert, 1999). P. amabilis is a wind-pollinated (Zanni & Ravazzi, 2007) and wind-dispersed species (Fordham & Spraker, 1997). The species grows on a variety of soils derived from acidic rock and is distributed in mixed-mesophytic and evergreen sclerophyllous broad-leaved forests (Wang, 1961; Farjon, 1990).

Currently, this species is ranked as at least ‘Vulnerable B2ab (iii, v) ver. 3.1’ and possibly the more threatened category ‘Endangered’ in the International Union for Conservation of Nature (IUCN) Red List of Threatened Species (Yang & Christian, 2013). Its population is severely fragmented, and the quality of the habitat and the population size continue to decline (Yang & Christian, 2013). To prevent deterioration, establishment of protected areas has been advocated by the IUCN (Yang & Christian, 2013). Compared to its current distribution, the extensive distribution of this species across the Northern Hemisphere was much wider during geological time (from the Cretaceous to the P1io-P1eistocene) (LePage & Basinger, 1995; Fig. S1). The sharp contraction suggests that the relict species P. amabilis has kept up with and survived past changes in climate. However, whether the ‘living fossil’ P. amabilis can cope with the challenge presented by the unprecedentedly rapid climate warming of the 21st century is a matter of concern. In addition, mast seeding—the synchronous intermittent production of large seed crops by populations (Kelly, 1994)—may aggravate the vulnerability of P. amabilis to climate change.

Reconstructing the range dynamics of species under past climate fluctuations is helpful for revealing their response mechanism to climate change. Clarification of a species’ response mechanism in combination with future distribution prediction can assist in developing effective conservation strategies (Petit, Hu & Dick, 2008; Désamoré et al., 2012). Ecological niche models (ENMs) link species occurrence records with environmental variables on multiple spatial and temporal scales to model past, present, and future distribution (Peterson et al., 2011). Such climate envelope models are important for revealing the past response of organisms to climate change, particularly for periods for which there is no fossil record. Overall, this tool has been regarded as an effective approach for reliably assessing the potential effects of future climate change on the fate of organisms (e.g., Huntley et al., 2004; Rodríguez-Sánchez & Arroyo, 2008; Elith & Leathwick, 2009).

Niche conservatism is a key assumption underlying the application of ENMs. In the case of P. amabilis, the evolutionary history indicates evolutionary stasis for this species since at least the Eocene (LePage & Basinger, 1995). As P. amabilis is an extraordinary example of evolutionary stasis, its niche is considered to be evolutionarily conservative because evolutionary stasis is perceived as related to niche conservatism (Eldredge et al., 2005; Stigall, 2012). The inference of niche conservatism for P. amabilis is supported by the general consistency of its temperature requirement throughout geological time (Bai & Li, 2017). In addition, this concern can be somewhat alleviated by the fact that a very general pattern of niche conservatism among species has been rather broadly confirmed in a recent review (Peterson, 2011).

In this study, we use an ENM to model the range dynamics of P. amabilis across four temporal frameworks representing four extreme moments of climatic variability during the Quaternary: the LIG, the LGM, the present and the future (2080). Our overall aim is to hind-cast the response of P. amabilis to chronological climate change and ultimately to provide some meaningful information about future conservation strategies for this vulnerable species based on the hind-casted response mechanism to climate change and the future distribution prediction.

Materials & Methods

Natural occurrence data for P. amabilis were compiled from the Chinese Virtual Herbarium (http://www.cvh.ac.cn/), the Global Biodiversity Information Facility (http://www.gbif.org/), and references (Ge et al., 1998; Hao et al., 2000; Duan et al., 2012; Huang et al., 2016).

Bioclimatic layers at a resolution of 2.5 arc minutes for the present (representative of 1960–1990), the LIG (∼120–140 ka) and the LGM (∼21 ka) climate were downloaded from the WorldClim dataset (available at http://www.worldclim.org/). Present climate data were generated through interpolation of average monthly climate data from global weather stations (Hijmans et al., 2005). LGM climate data were based on general circulation model (GCM) simulations from the Community Climate System Model (CCSM; Kiehl & Gent, 2004), and LIG climate data were based on models from Otto-Bliesner et al. (2006).

Future climate data were provided by the CGIAR Research Program on Climate Change, Agriculture and Food Security (CCAFS) (available at http://www.ccafs-climate.org). These climate data were generated by GCMs under a set of emissions scenarios. By the end of 2012, the closest emissions scenarios resulting from the Intergovernmental Panel on Climate Change (IPCC) process to the observed emission trends were the Special Report on Emissions Scenarios (SRES) A1B used in the IPCC Fourth Assessment Report and the Representative Concentration Pathways (RCPs) 8.5 used in the IPCC Fifth Assessment Report (Peters et al., 2012). The SRES A1B scenario describes a future of very rapid economic growth, a balance between the use of fossil fuels and non-fossil fuels and moderate human population growth (Intergovernmental Panel on Climate Change, 2007); the RCP 8.5 scenario depicts a world characterized by an atmospheric CO2 concentration that continues to increase at current rates (Intergovernmental Panel on Climate Change, 2013). For this study, we chose four types of future (2080) climate layers generated from simulations using a set of GCMs under the above two emission scenarios: UK Meteorological Office (UKMO) Hadley Centre Coupled Model, version 3 (HadCM3) (Gordon et al., 2000; Pope et al., 2000), under SRES A1B; Met Office Hadley Centre (MOHC) Hadley Centre Global Environmental Model, version 2 (HadGEM2-ES (Earth System)) (Jones et al., 2011), under RCP 8.5; Canadian Centre for Climate Modelling and Analysis (CCCma) third-generation Coupled Global Climate Model with T63 spectral resolution (CGCM3.1-T63) (Flato, 2005), under SRES A1B; and Meteorological Research Institute (MRI) Coupled Global Climate Model, version 3 (CGCM3) (Yukimoto et al., 2012), under RCP 8.5.

The maximum entropy (MaxEnt) model, based upon the maximum entropy principle and using ‘presence-only’ species data (Phillips, Anderson & Schapire, 2006), has excellent predictive performance for threatened or range-restricted species (Elith et al., 2006; Hernandez et al., 2006; Hijmans & Graham, 2006; Pearson et al., 2007). We employed MaxEnt 3.4.1 for modelling. The responses of multivariate nonlinear models based on highly correlated climate variables can result in model overfitting (Morlini, 2006), which will also occur when such models are applied to data outside the training conditions (Graham, 2003; Morlini, 2006). To prevent multicollinearity and increase transferability effectiveness, we removed all environmental predictors with high pairwise correlation (Pearson’s correlation > 0.85, Table S1). Among highly correlated variables, we selected those with more direct physiologically roles in limiting the survival and reproduction of P. amabilis. We also deleted climate variables with a relatively low percentage contribution to the model performance. The contribution of each climate variable was assessed by the Jackknife procedure in MaxEnt (Fig. S2). These operations created a model of better performance with a balance between an underfitted model with few parameters and an overfitted model with too many correlated explanatory variables (Burnham & Anderson, 2002). The final set of selected bioclimatic variables included mean annual temperature (MAT, Bio1), temperature seasonality (TS; standard deviation of monthly mean temperature value, Bio4), min temperature of coldest month (MTCM, Bio6), mean temperature of driest quarter (MTDQ, Bio9), annual precipitation (AP, Bio12), precipitation of driest month (PDM, Bio14), precipitation of wettest quarter (PWeQ, Bio16) and precipitation of warmest quarter (PWaQ, Bio18).

ENMs assume that species are in equilibrium with their environments (Guisan & Zimmermann, 2000), i.e., species occur in all climatically suitable regions while being absent from all unsuitable ones (Hutchinson, 1957). In the process of investigating species occurrence data, however, some areas in a landscape were sampled more intensively than others; this sampling bias will lead to a lack of equilibrium of species distributions with climate. To reduce the effect of sampling bias, the approach of spatial filtering of a species occurrence dataset was applied in this study (Boria et al., 2014). We spatially filtered the distribution data of P. amabilis to obtain the maximum number of localities 10 km apart. As a result, 49 occurrence localities (Table S2) were rarefied to 47 points (Table S3). To minimize the effect of sampling bias, we also used a bias file representing a Gaussian kernel density of the species occurrence localities sampled within a 60-km search radius. The bias file upweights presence-only data points with fewer neighbours in a geographic space (Elith et al., 2011). The species occurrence data and climate layers were both projected to the Asia north equidistant conic projection in ArcMap.

We chose area under the receiver operating characteristic curves (AUCs) to assess the predictive performance of the ENMs (Fielding & Bell, 1997). The AUC statistic is bounded from 0.5 to 1.0, in which 0.5 indicates a random prediction (useless model) and 1.0 a perfect model prediction of presence versus absence; the closer AUC is to 1, the better the predictive accuracy of the model. Each model was run 10 times, with 75% of the species occurrence data selected for model training and 25% for model testing. To obtain binary predictions of the climate suitability of ENMs, MaxEnt’s logistic probability of occurrence output was converted to a binary mode (presence-absence output) using the maximum training sensitivity plus specificity logistic (MTSS) threshold and the 10 percentile training presence logistic (10% TP) threshold. By maximizing the proportions of actual positives and negatives that are correctly identified, prediction based on the MTSS threshold represents the most accurate forecast of presence/absence (Liu et al., 2005; Jiménez-Valverde & Lobo, 2007). The prediction based on the 10% TP threshold represents the core of species ranges by excluding the 10% of training localities with lowest prediction in the modelling (Morueta-Holme, Fløjgaard & Svenning, 2010; Anderson & Gonzalez, 2011).

To explore the distributional change between the two ENMs during each climate transition (e.g., from the LIG to the LGM, from the LGM to the current period, and from the present to 2080), we reduced the distribution to a single central point (known as a centroid) and created a vector depicting the magnitude and direction of the predicted change. Furthermore, to reveal potential migration in detail, we applied migration vector analysis. First, we calculated the geographic centroids of the ranges for every 60 × 60 km2 for each period and then determined the centroid in the second time interval nearest to the centroid in the first time interval (i.e., LGM to LIG, present to LGM, and future to present); finally, we evaluated the potential migration from one period to the next.

To determine the variable with greatest influence for the model prediction in each grid cell, we performed limiting factor analysis. For any given point, the limiting factor is the variable with a value change at that point that results in the greatest change in the predicted probability of species occurrence (Elith, Kearney & Phillips, 2010; Elith et al., 2011). We focused on the limiting climatic factors affecting the contraction and expansion of climate suitability. The algorithm for the limiting factor analysis was programmed into MaxEnt, and the ‘density.tools.LimitingFactor’ command was used.

Results

The observed AUC values of 0.9802 (training data) and 0.9744 (testing data) indicate a good discrimination of the models between absence and presence cells, and the relatively high AUC values suggest that the species distributions were well predicted by climate. The MTSS and 10% TP thresholds were calculated to be 0.3048 and 0.3842, respectively.

It is noteworthy that the predicted area based on the MTSS threshold is very similar to that based on the 10% TP threshold (Figs. 1A–1D). The high similarity implies a consistence in the trend of distributional change over time between the predictions based on the two types of thresholds. Therefore, investigation of the predicted areas was based on binary ENMs generated by MTSS and 10% TP thresholds; however, assessment of the dynamic change of distribution over time was based only on the chosen MTSS threshold. Besides, compared to the present distribution of P. amabilis, its future distributions predicted using four types of climate layers all indicate an expansion northward and southern contraction, though the areas of the expansion and contraction differ (Fig. S3). Given the consistent trend of change and the focus of this study to explore the distributional change and potential migration routes of species, for conciseness, we choose one of the four future prediction for elaboration below: the prediction based on the HadCM3 simulation under SRES A1B.

Figure 1 Maps of predicted climate suitability forPseudolarix amabilis across time stages.

The Last Interglacial (A), the Last Glacial Maximum (B), the present (C), and 2080 (D). The spatial data was freely downloaded from http://www.diva-gis.org/Data, the base map was generated by ArcGIS v.9.3 (http://www.esri.com/software/arcgis/arcgis-for-desktop).

The current potential distribution of P. amabilis involves three main disjunct districts (Fig. 1C): southeast China, where the actual population exists; and the southern frontier regions of Japan and the Korean peninsula, where extant populations are absent. The ENM projections for the other three periods (LIG, LGM and 2080) depict potential distributions with altered locations. The modelled potential climate suitability during the LIG shows four main disjunct regions, including southeast China, the southern frontier regions of the Himalayas and the southern frontier regions of Japan and the Korean Peninsula (Fig. 1A). During the LGM, fragmented distribution in southeast China is revealed (Fig. 1B), whereas the central-eastern regions of China, most parts of the Korean Peninsula and southern Japan are included in the future (Fig. 1D).

Contraction and expansion occur during each transition, and the resulting range varies significantly through time (Fig. 2A, Table 1). The ratio of the area between adjacent moments indicates a dramatic shrinkage from the LIG to the LGM, followed by significant expansion from the LGM to the current time and moderate expansion from the current period to the future (Fig. 2B, Table 1). The pattern of climate suitability change is dominated by contraction from the LIG to the LGM (Fig. 3A, Table 2), and by expansion since the LGM (Figs. 3C and 3E, Table 2).

Figure 2 The predicted area during four time intervals (A) and the area ratios between adjacent time periods (B).

Table 1 The predicted area during the four time periods and their ratios through time.

	Threshold	LIG	LGM	Present	2080	
Area (104 km2)	MTSS	116.5	33.8	122.2	179.6	
10% TP	100.8	12.6	88.9	151.5	
		LGM/LIG	Present/LGM	2080/Present	
Ratio of area (%)	MTSS	29.0	362.0	146.9	
10% TP	12.5	703.7	170.3	

Figure 3 Migration of Pseudolarix amabilis. over climate transitions.

Distributional change (A, C, E) and migration distance and direction (B, D, F) of Pseudolarix amabilis over climate transitions: from the LIG to the LGM (A, B), from the LGM to the present (C, D), and from the present to 2080 (E, F). The large arrows in a, c and e indicate the overall migration routes; the small arrows in b, d and f indicate the potential migration in detail. The spatial data was freely downloaded from http://www.diva-gis.org/Data, the base map was generated by ArcGIS v.9.3 (http://www.esri.com/software/arcgis/arcgis-for-desktop).

With regard to distributional change, overall migration route analysis shows migration southeast from the LIG to the LGM (Fig. 3A), followed by migration northeast from the LGM to the present (Fig. 3C) and migration northward from the present to 2080 (Fig. 3E). From the LIG to the LGM, the expansion occurred mainly in the Sichuan Basin, with a source population from the most southeastern regions of China migrating northwest (Fig. 3B). From the LGM to the present, the expansion occurred in southeast China, with a relatively short migration distance (Fig. 3D). Northward expansion is the dominant tendency from the present to 2080 (Fig. 3F).

Table 2 The change in predicted area based on the MTSS threshold through time.

	From LIG to LGM	From LGM to Present	From Present to 2080	
Constriction (104 km2)	97.0	13.1	43.6	
[ratio (%)a]	[83.2]	[38.7]	35.6	
No change (104 km2)	19.5	20.6	75.9	
[ratio (%)a]	16.7	61.0	62.1	
Expansion (104 km2)	14.2	101.6	103.7	
[ratio (%)b]	[42.0]	[83.2]	[57.8]	
Notes.

a indicates the percentage of changed area relative to the previous period (LIG, LGM and Present).

b indicates the percentage of changed area relative to the latter period (LGM, Present and 2080).

Limiting factors analysis indicates that the main climate variables influencing the model prediction for the change in climate suitability are temperature seasonality (TS), min temperature of coldest month (MTCM), mean temperature of driest quarter (MTDQ), annual precipitation (AP) and precipitation of driest month (PDM) (Figs. 4A–4C, Table 3, Fig. S4). The changes in the five variables set them within or outside the range of the physiological tolerances of P. amabilis (Fig. S5) and result in distribution expansion or contraction.

Figure 4 Main limiting climatic factors for the distributional changes of Pseudolarix amabilis.

The main limiting climatic factors for the distributional changes (contraction and expansion) of Pseudolarix amabilis and the amplitude of change of each main limiting factor over climate transitions: from the LIG to the LGM (A), from the LGM to the present (B), and from the present to 2080 (C). The change amplitude of the limiting factor was obtained by subtracting the previous values from the latter values, i.e., LGM minus LIG (A), present minus LGM (B), and 2080 minus present (C). Contraction is outlined in red; expansion is outlined in black. Bio4: temperature seasonality, Bio6: min temperature of coldest month (°C), Bio9: mean temperature of driest quarter (°C), Bio12: annual precipitation (mm), Bio14: precipitation of driest month (mm). The spatial data was freely downloaded from http://www.diva-gis.org/Data, the base map was generated by ArcGIS v.9.3 (http://www.esri.com/software/arcgis/arcgis-for-desktop).

Discussion

Formation of glacial refugia

Decreasing temperatures, especially winter temperatures, dominated the trend of climate change from the LIG to the LGM. As expected, the marked decrease in winter temperature caused min temperature of coldest month (MTCM) to be the main limiting factor accounting for the northern contraction of P. amabilis (Fig. 4A), which is a thermophilic tree according to its physiological tolerance of MTCM (Fig. S5). Low temperatures affect the survival of plants by impacting their transition from vegetative to reproductive development (Gallagher, 1986) as well as their assimilation of soil water and nutrients for cell division, differentiation and tissue growth (Thuiller et al., 2005). Extremely cold winters can even result in frost kill (Pearson et al., 2002). Therefore, the significance of minimum temperatures in determining the world distributions of species, especially the northern boundary of their ranges, has been long recognized (Raison et al., 1979; Woodward, 1987; Ashcroft, Chisholm & French, 2008). Moreover, a recent evaluation via ENMs of environmental factors affecting species distributions indicated that winter minimum temperature contributes the most to model predictions (Ashcroft, French & Chisholm, 2011). In contrast, temperature seasonality (TS) and annual precipitation (AP) were important factors for the observed southern contraction of P. amabilis (Fig. 4A). Notably, the change in TS was a main factor leading to the expansion in the Sichuan Basin (Fig. 4A). The contrasting role played by the same climate variable against the same climate background may result from heterogeneity among regional climate conditions and differences in the variation of climate variables.

Table 3 The percentage of main limiting factors accounting for the variation of climate suitability through time.

	From LIG to LGM	From LGM to Present	From Present to 2080	
Contraction	Bio6	45.1%	Bio4	66.9%	Bio14	43.2%	
Bio4	22.7%	Bio12	33.1%	Bio4	40.8%	
Bio12	22.1%					
Expansion	Bio4	46.7%	Bio12	30.1%	Bio12	60.7%	
Bio9	40.6%	Bio4	28.3%			
Notes.

Bio4 temperature seasonality (standard deviation)

Bio6 min temperature of coldest month (°C)

Bio9 mean temperature of driest quarter (°C)

Bio12 annual precipitation (mm)

Bio14 precipitation of driest month (mm)

Rapid climate change, especially the rapid cooling and warming that occur at the beginning and end of glaciation cycles, is always accompanied by the extinction of populations that fail to track climate shift or adapt to new conditions (e.g., Hofreiter & Stewart, 2009; Loarie et al., 2009; Corlett & Westcott, 2013). Therefore, aggravated by the risk of failure of tracking and adaptation, the drastic range shrinkage of P. amabilis may have once placed this species near extinction.

The threat of extinction makes the presence of refugia meaningful. As the sole area of distribution expansion from the LIG to the LGM, the Sichuan Basin acted as an important refuge for P. amabilis, in addition to scattered refugia in southeast China (Fig. 3A). Fossil records of glacial periods have been used to confirm glacial refugia of organisms, though regrettably, no fossils of P. amabilis have been found in the above refugia.

Post-LGM colonization

Affected by the main limiting factors of AP and TS (Fig. 4B), expansion dominated the change in distribution from the LGM to the present, but contraction also occurred on a small scale (Fig. 3B). The present-day ranges of living P. amabilis generally agree with its predicted climate suitability (Fig. 3D), suggesting that climate is the main determinant in constraining plant species ranges, which is consistent with many other studies (e.g., Prentice, Bartlein & Webb, 1991; Pearson & Dawson, 2003; Heikkilä, Fontana & Seppä, 2009). However, the absence of an extant population of P. amabilis from current climatically suitable areas is also noticeable (Figs. 3C and 3D), indicating that other than climatic factors may constrain the ability of this species to colonize its potential range. Constraints by nonclimatic factors have also been demonstrated by the postglacial expansion of European tree species (Svenning & Skov, 2004; Normand et al., 2011).

The absence of extant species from climatically suitable areas may result from time-lagged migration or the exclusion of species from established colonization (Normand et al., 2011). The migration performance of a species is related to its intrinsic dispersal ability and extrinsic influencing factors, such as the locations of ice age refugia, geographical barriers, habitat fragmentation and competition with established vegetation (Davis, 1986; Prentice, Bartlein & Webb, 1991; Svenning & Skov, 2004). The factors that drive a species out of an established colonization area include the local edaphic conditions, biotic interactions and human deforestation. The factors contributing to the absence of P. amabilis from climatically suitable areas depend on the specific geographic position.

In southeast China, multiple factors may be responsible for the absence of P. amabilis in climatically suitable areas. For example, the relatively high mountains southeast of the Sichuan Basin and north of Guangxi Province (Fig. 3D) may have blocked the tree from moving into the surrounding areas. This block led to its incomplete expansion and its absence from the southwestern part of the potential climatically suitable area. In contrast, in areas that are not blocked by high mountains, exclusion of populations from established colonization may be the main reason for the absence of P. amabilis, such as in Jiangxi Province, the eastern part of Hubei Province and the southern part of Anhui Province (Fig. 3D). Another notable absence occurred in the northeastern regions of the predicted climatically suitable areas, such as Jiangsu Province (Fig. 3D). The migration distance required to fully cover climatically suitable areas is as great as ∼300–480 km, and the absence of P. amabilis may indicate that the actual distance migrated was shorter than the theoretically required value, revealing a limited dispersal ability. Although certain factors played a leading role in the absence of P. amabilis from specific regions, the contributions of other abovementioned factors should not be overlooked.

By contrast, the reasons for the absence of this species in Korea and Japan are different from those in southeast China. The location of LGM refugia can impact the post-LGM colonization of species (Firbas, 1949), principally by affecting the expansion of species to current climatically suitable areas (Normand et al., 2011). The absence of glacial refugia in Korea and Japan led to the lack of a resource population for expansion. At the same time, the limited long-distance dispersal ability of P. amabilis and separation by climatically unsuitable areas and the ocean resulted in the impossibility of colonization from the refugia in southeast China. Consequently, the failure of post-LGM colonization led to the absence of living P. amabilis in Korea and Japan, despite a suitable climate in those regions.

Overall, the postglacial range dynamics of P. amabilis suggest that the present climate strongly shaped the current distribution of this species; however, other forces, such as its limited long-distance dispersal ability, geographical barriers or human influence, constrained its ability to completely fill potential climatically suitable areas. Our results support the view that although climate exerts a dominant control over the natural distribution of species on a regional to global scale, non-climatic factors play important supplementary roles at the local level (Svenning & Skov, 2004; Normand et al., 2011).

Perspectives for the future

In response to future climate change, the potential climate suitability of P. amabilis will move northward, resulting from contraction south and expansion north (Fig. 3E). The contraction is controlled by changes in the main climate variables precipitation of driest month (PDM) and TS, whereas the expansion is controlled mainly by changes in AP (Fig. 4C). This type of latitudinal shift, via range shifts from lower to higher latitudes, has been regarded as the widespread response of species to future climate change (Parmesan & Yohe, 2003; Hof et al., 2011).

The area of expansion is greater than that of contraction, resulting in a larger area of climate suitability in the future than in the present (Fig. 3E). However, the increase in this area does not substantially relieve concerns about the future destiny of P. amabilis. This is because the expanded climatically suitable areas may not be completely filled by dispersal and because the areas of southern contraction will become climatically unsuitable.

The incomplete range filling of P. amabilis may be related to its limited accessibility to climate suitability, as shown in its response to post-LGM climate change. As portrayed in its post-LGM colonization, geographic barriers, landscape modifications and low migration ability may continually affect the expansion of P. amabilis. For example, the Daba Mountains may prevent the northward expansion of the population from the Sichuan Basin (Fig. 3F). Although no high mountains are present, landscape modifications, such as land-use changes and concomitant habitat destruction, degradation and fragmentation in Henan and Shandong (Fig. 3F), may disrupt dispersal processes (Haila, 2002; Fazey, Fischer & Lindenmayer, 2005; Fischer & Lindenmayer, 2007). In contrast, the limitation of the low migration rate of P. amabilis appears particularly prominent against the background of the unprecedented rate of ongoing climate change. This unprecedented climate change necessitates species dispersal that is rapid enough to match the climate shifts (Huntley, 1997; Hoegh-Guldberg et al., 2008). Theoretically, to keep up with climate change in the coming decades, P. amabilis must move from its dispersal source in China up to an elevation of 750 km by the end of 2080, with most distances being greater than 100 km. (Because there is no living P. amabilis in Korea and Japan, expansion there in the future is not realistic.) The migration rate necessary to achieve such great distances in the coming decades is greater than the inferred rate of range shifts of 300 to 500 km per century that is required for plants to track climate change in the 21st century (Davis & Shaw, 2001). However, the actual migration rate of P. amabilis, as deduced from the process of its post-LGM colonization, may be much lower. Without knowledge of the actual migration rate of P. amabilis, the commonly observed past migration rates of trees of 20 to 40 km per century (Davis, 1986; Davis & Shaw, 2001) can provide a reference. Therefore, the lower migration rate of P. amabilis relative to the climate-change velocity will also likely lead to its failure to fully colonize the future climatically suitable areas.

By comparison, the population confined to the southern contraction regions will face a different challenge: struggling with the burden of the upcoming unsuitable climate. To avoid extinction, adaptation to new climate conditions is an alternative in addition to migration, and the adaptation rate must be in equilibrium with the rate of climate change (Dawson et al., 2011). Regardless, climate change commonly overwhelms the adaptation of species (Davis, Shaw & Etterson, 2005; Petit, Hu & Dick, 2008) and almost certainly will in the future because of the unprecedentedly rapid rate (Davis & Shaw, 2001; Jump & Penuelas, 2005). Because of its evolutionary stasis for at least 56 million years ago, there is a slight possibility of niche evolution for P. amabilis for adaptation in the coming decades (LePage & Basinger, 1995). Moreover, although some behavioural and/or evolutionary adaptations may occur, the adaptation rate will likely be too slow to match the unprecedented rapidity of climate change, largely due to the slow reproductive rate resulting from the mast seeding and long lifespan of the species. Consequently, the high rate of niche evolution required for a high adaptation rate stands in stark contrast to the supposed evolutionary stasis in P. amabilis’ niches. This suggests that the population of P. amabilis confined to the southern contraction areas may not adapt to the coming new climate.

Given the limited accessibility of certain species to climatically suitable areas and their inability to adapt to new climate conditions in situ, assisted migration has been suggested as a supplementary means of conservation (Hunter, 2007; Hoegh-Guldberg et al., 2008). The feasibility of applying this method to P. amabilis has been confirmed by the success of current cultivated introductions in a variety of sites, even in climatically unsuitable areas, such as the National Forest Park of Yaoxiang in Shandong Province and Xiaolongshan Botanical Garden in Ganshu Province. However, one major concern associated with assisted migration is the potential for disrupting the native ecological balance at the target sites (McLachlan, Hellmann & Schwartz, 2007; Hoegh-Guldberg et al., 2008). Given that most major ecological invasions have occurred via continent-to-continent and continent-to-island translocations (e.g., Weber, Sun & Li, 2008; Alexander et al., 2009), translocations of P. amabilis within east China are unlikely to create devastating negative effects. In addition, the mast seeding of P. amabilis with an approximate 5-yr cycle and its limited migration ability suggest a very low possibility that it will exhibit invasive tendencies in introduction areas. Thus, assisted migration is likely to be an effective conservation strategy.

In addition to the latitudinal shift in distribution, a shift towards higher elevations is suggested as an additional response of P. amabilis. This elevational shift is predicted to mainly occur in unchanged climatically suitable areas. Compared to latitudinal migration, unassisted elevational shift may be feasible for P. amabilis, as the migration distances to climatically suitable areas are relatively short. Nonetheless, to increase the probability of success in elevational shift, assisted migration in elevational direction should also be considered.

Conclusions

In summary, in common with the general responses of species to climate change but with individual pattern, P. amabilis responds to glacial-interglacial cycles with high sensitivity, supporting the view that restricted-range species are sensitive to climate change (Sandel et al., 2011). The combination of investigating the response mechanism of P. amabilis to past climate change and predicting future climate suitability is beneficial for devising an effective conservation strategy. Our findings highlight the importance of combining a historical perspective with future predictions to develop a global conservation planning strategy for organisms in a changing world.

Uncertainties related to our model should be kept in mind when interpreting the results of this study. One important uncertainty derives from the fact that non-climatic factors, such as soil conditions, could not be integrated into our modelling because of the lack of sufficient data so far. In addition, many other ecological and evolutionary processes, such as biological interactions and interactions between the functional traits of an organism and its habitat, will also affect the distribution of species (Kearney & Porter, 2009; Fordham et al., 2012). These constraints are being addressed by some mechanistic modelling approaches. Compared to the MaxEnt-style correlative/statistical model, which statistically links spatial data to species distribution records, mechanistic models incorporate mechanistic links between the functional traits of organisms and their environments (e.g., Renton, Shackelford & Standish, 2012; Tomlinson et al., 2017). These two types of models have both strengths and weaknesses (Kearney & Porter, 2009; Kearney, Wintle & Porter, 2010). For poorly studied taxa with a paucity of knowledge about the physiological constraints on their survival and reproduction, such as P. amabilis, MaxEnt-style correlative/statistical models are a better choice. We will never be able to reconstruct the past and predict the future with accuracy, but we need a strategy for utilizing existing knowledge to reveal the likely effects of climate on species survival. Optimistically, we hold the opinion that as more data become available, ENMs will generate more realistic simulations and provide a solid basis on which to draw a more practical conservation strategy.

Supplemental Information

Figure S1 The fossil and extant occurrence of Pseudolarix amabilis

The spatial data was freely downloaded from http://www.diva-gis.org/Data, the base map was generated by ArcGIS v.9.3 (http://www.esri.com/software/arcgis/arcgis-for-desktop).

Click here for additional data file.

Figure S2 The contribution of all 19 climate variables to model prediction

The Jackknife test for evaluating the contribution of all 19 climate variables from the WorldClim dataset for model training (A) and testing (B). Bio1: annual mean temperature, Bio2: mean diurnal range (mean of monthly (max temp - min temp)), Bio3: isothermality (Bio2 / Bio7) (* 100), Bio4: temperature seasonality (standard deviation *100), Bio5: max temperature of warmest month, Bio6: min temperature of coldest month, Bio7: temperature annual range (Bio5–Bio6), Bio8: mean temperature of wettest quarter, Bio9: mean temperature of driest quarter, Bio10: mean temperature of warmest quarter, Bio11: mean temperature of coldest quarter, Bio12: annual precipitation, Bio13: precipitation of wettest month, Bio14: precipitation of driest month, Bio15: precipitation seasonality (coefficient of variation), Bio16: precipitation of wettest quarter, Bio17: precipitation of driest quarter, Bio18: precipitation of warmest quarter, Bio19: precipitation of coldest quarter.

Click here for additional data file.

Figure S3 Four types of predictions of climate suitability for Pseudolarix amabilis in the future (2080)

Four types of predictions of climate suitability for Pseudolarix amabilis in the future (2080) using climate layers generated by the following: UK Meteorological Office (UKMO) Hadley Centre Coupled Model, version 3 (HadCM3), under SRES A1B (A); Met Office Hadley Centre (MOHC) Hadley Centre Global Environmental Model, version 2 (HadGEM2-ES (Earth System)), under RCP 8.5 (B); Canadian Centre for Climate Modelling and Analysis (CCCma) third-generation Coupled Global Climate Model with T63 spectral resolution (CGCM3.1-T63) under SRES A1B (C); and Meteorological Research Institute (MRI) Coupled Global Climate Model, version 3 (CGCM3), under RCP 8.5 (D). The spatial data was freely downloaded from http://www.diva-gis.org/Data, the base map was generated by ArcGIS v.9.3 (http://www.esri.com/software/arcgis/arcgis-for-desktop).

Click here for additional data file.

Figure S4 All the limiting climatic factors responsible for the distributional change of Pseudolarix amabilis

All the limiting climatic factors responsible for the contraction and expansion of distribution of Pseudolarix amabilis over climate transitions: from the LIG to the LGM (A), from the LGM to the present (B), and from the present to 2080 (C). Bio1: annual mean temperature, Bio4: temperature seasonality (standard deviation *100), Bio6: min temperature of coldest month, Bio9: mean temperature of driest quarter, Bio12: annual precipitation, Bio14: precipitation of driest month, Bio16: precipitation of wettest quarter, Bio18: precipitation of warmest quarter. The spatial data was freely downloaded from http://www.diva-gis.org/Data, the base map was generated by ArcGIS v.9.3 (http://www.esri.com/software/arcgis/arcgis-for-desktop).

Click here for additional data file.

Figure S5 The range of the physiological tolerances of Pseudolarix amabilis

Bio4: temperature seasonality (A), Bio6: min temperature of coldest month (°C) (B), Bio9: mean temperature of driest quarter (°C) (C), Bio12: annual precipitation (mm) (D), Bio14: precipitation of driest month (mm) (E).

Click here for additional data file.

Table S1 The correlation among 19 climate variables

Click here for additional data file.

Table S2 The natural occurrence data of Pseudolarix amabilis

Click here for additional data file.

Table S3 The filter ed occurrence data of Pseudolarix amabilis with 10-km filtering distance

Click here for additional data file.

We are grateful to the Chinese Virtual Herbarium and the Global Biodiversity Information Facility for providing the distributional data of Pseudolarix amabilis. We are also thankful to the editor and three referees for their constructive comments and Dr. Huijie Qiao, Institute of Zoology, Chinese Academy of Sciences, Beijing, China, for his suggestions on the revision of this study.

Additional Information and Declarations

Competing Interests

Author Contributions

Data Availability

The authors declare there are no competing interests.

Yunjun Bai performed the experiments, analyzed the data, contributed reagents/materials/analysis tools, wrote the paper, prepared figures and/or tables.

Xueping Wei performed the experiments, analyzed the data, wrote the paper, prepared figures and/or tables.

Xiaoqiang Li conceived and designed the experiments, analyzed the data, wrote the paper, reviewed drafts of the paper.

The following information was supplied regarding data availability:

The raw data is provided in Figs. S1–S5 and Tables S1–S3.

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
