# Peer review of "Distributional dynamics of a vulnerable species in response to past and future climate change: a window for conservation prospects"

_PeerJ, doi:10.7717/peerj.4287_

## Round 0.1 · original submission · Major Revisions

Dear Colleagues,

I have 3 very detailed and honest reviews in my hands. These, if you follow their advice, would improve your paper considerably. Please, disregard the musings of one of the referees about the originality of the approach as this is not targeted by PeerJ policies. Solid science, presented in lucid way and correct language is what matters.

On that note: Please, take each of the concerns rose by the referees (see the attached referees' reports) seriously and try to address them;
There are several major issues I want you to focus to get it right:

1) at least 2 referees require more information on the organism you have studied and its ecology - a brief account would be helpful;
2) you have to explain the choice of your climate prediction model and provide further details as requested by the referees;
3) you are requested to streamline your Introduction - please focus on 3 major issues: motivation (with brief literature overview); identification of the gap in knowledge; aims;
4) Discussion also needs a major overhaul; as a rule the Discussion should reflect Aims --> that should be reflected in the structure of Results --> that should be than addressed in Discussion; the Discussion needs major trimming;
5) As often, language (style especially) is a problem. Please, do involve an English-native speaker (preferably biologist) to help you to streamline your language. It is quite convoluted in places and really needs major attention.

Sincerely
Prof Ladislav Mucina, Academic Editor

Reviewer 1 ·

Basic reporting

Initially I was a bit concerned about the use of English but this was quickly dispelled. The article is generally well written however I request that the earlier sections, up-and-till the Results, be re-checked for use of English, particularly sentence construction.

Experimental design

The reader may not be familiar with Pseudolarix amabilis hence it is important to provide a bit more useful background but relevant information. State that it is a tree, occurs family Pinaceae, and provide a few additional plant traits such as it has cones with winged seeds, pollination by wind, longevity, etc.

Provide its full IUCN red list status.

Sentence in Lines 145-147. Please reword ending as approach is not clearly articulated.

Figure 3 (A). The arrow direction should be reversed as it currently shows migration from expansion areas to contraction/no change areas. (B) is correct.

"Figure 4. Main limiting climatic factors for the distribution ..” add word “climatic”

Validity of the findings

Lines 408 to 411. I am uncomfortable with this statement. The model shows sensitivity to climate change and not necessarily the species, given the limitations discussed.

Additional comments

There is no mention of the likelihood that some populations occurring during the LIG and LGM periods could have gone extinct and therefore their locations and influence on Maxent model could not be included (and therefore the models may be slightly more conservative, failing to include extinct populations).

·

Basic reporting

Overall the paper seems fairly sound, and should eventually be published I believe. It uses well established methods to predict past, present and future suitable climate areas for one particular plant species It has various issues that should be addressed before it is published, as detailed below.


Literature references and sufficient field background/context are provided in most places, with some exceptions noted below.

The structure of the paper is ok, but some revision of Discussion is required (Discussion too long, first paragraph should focus on your results, contains some material that should be in Results, see below for details).

I would say the paper just represents an appropriate 'unit of publication'. SDM studies like this are becoming so easy to do, with the general availability of data on species presence and climatic variables. It would seem relatively easy to do the same study done here for another species, and another, and another and so on. I wonder whether each one would really deserve its own publication? I think studies that focus on several species at once are already becoming the norm (eg. Yates et al and Hamer et al below). Perhaps these could be referenced as examples of what such studies could try to encompass in future? Nonetheless, I think this study just meets the requirements for an appropriate 'unit of publication'.

Yates, C.J., McNeill, A., Elith, J., Midgley, G.F., 2010. Assessing the impacts of climate change and land transformation on Banksia in the South West Australian Floristic Region. Diversity and Distributions 16, 187–201. https://doi.org/10.1111/j.1472-4642.2009.00623.x

Hamer, J.J., Veneklaas, E.J., Poot, P., Mokany, K., Renton, M., 2015. Shallow environmental gradients put inland species at risk: Insights and implications from predicting future distributions of Eucalyptus species in South Western Australia. Austral Ecology 40, 923–932. https://doi.org/10.1111/aec.12274



The English language should be improved to ensure that an international audience can clearly understand your text. In many places the choice of words, or tense or expression is a real barrier to understanding the science.


For example, in the abstract:

“To avoid potential biodiversity loss, biological conservation is
significant,” should be “To avoid potential biodiversity loss, biological conservation is
important,”

“While future modelling discovers a significant migration north of the potential climatically suitable areas of this species relative to current ones.” is not a sentence.

Lines 65-66

121 ‘literatures’ is not correct

The tense used seems incorrect at times. For example, the present tense is used in Materials and Methods for things that were clearly done in the past. This has real consequences for understanding the meaning of the text. These is sometimes an inappropriate mix of tenses.

Further examples are provided in the general comments below.

Experimental design

The study seems to follow a very established standard procedure for MaxEnt species distribution modelling and uses well-established software and methods. There is little that is original in the approach, beyond applying the established methods to a new species, but the flip-side of that is that there is little to criticize since the methods are so well established. The authors do also acknowledge some of the limitations of this kind of approach in the Discussion, which is good to see, but this should be better referenced, including references to more mechanistic approaches that are trying to overcome some of these limitations (see below for more details).

Validity of the findings

There is very little detail given on the climate projections used, and yet obviously all the results depend intimately on the climate assumptions used. The climate layers for the past climates and especially the one for the future climate must be very suspect, in terms of accuracy. This is somewhat acknowledged in the Discussion, but I would like to see some more details in the Methods as well, on how these climate layers are generated and how confident we can be in them (see details below).

Additional comments

The use of the word ‘window’ in the title seems strange to me. What does it mean??

Abstract, line27 “The response mechanism revealed,” It is unclear what the response mechanism is as it has not been mentioned previously in the abstract.

98-99 not clear to me how reconstructing range dynamics can reveal actual mechanisms… can you explain or provide a reference?

110 you cannot say that MaxEnt always has excellent predictive performance. It will always depend on the data, whatever the modelling approach used.

98-112 Overall this paragraph is not convincing. Maybe this is because of the references to response mechanisms. MaxEnt and most climate envelope methods are NOT mechanistic… they are purely statistical/descriptive. 101-112 is a reasonable summary of this kind of modelling, but its justification at the start is not convincing to me.

115 Following from the previous paragraph, the overall aim is not clear. ‘revealing the response’ – ‘revealing’ sounds like you are finding the actual response, but rather you are ‘predicting’ or estimating or ‘modelling or foracsting/hindcasting’ I think. “provide a theoretical foundation for future conservation strategies” – this sounds too vage to me to be a real scientific aim, or at least it needs more detail on how the distribution predictions will help future conservation strategies

124: I think some more information is needed on these climate layers, especially the past and future ones. How are they generated? How confident are we in them? For the future one, I think there are many different models and methods and emission scenarios that can be used to generate such layers… which were used in this case? How do your results depend on these choices and would conclusions be different if different climate models and emission scenarios were assumed?


127 – model overfitting is model overfitting… I would say that it cannot be exacerbated in certain conditions. However, the impacts of overfitting will be seen when the data is applied to new conditions.

129 – you don’t remove all environmental predictors with high pairwise correlation I think… you remove all except one??

131 – what is “the physiological role of P. amabilis”?? I think the word choice is wrong.

131-132: what is a low percentage contribution? How low is low?

135: “too many correlated parameters” – no, it is not the parameters that are correlated, it is the explanatory variables, right?

144: These two sentences seem unconnected, so I do not understand why ‘however’ is used for contrast here. How is the first sentence related to the rest of the paragraph?

149: why is it a bias FILE? Is file the right word choice?

151: what layers? ‘layers’ hasn’t been mentioned in this paragraph, or the one before. Is this something to do with the bias ‘file’?

153: the use of present tense here makes it sound like the authors are claiming this is the only way to assess predictive performance, but it is definitely NOT. I think they mean that this is how they chose to assess it??

159: and here past tense its used (was). The mix of past and present tense is very confusing, as it strongly impacts the meaning, as per the example above.

165: “by excluding the 10% most extreme data points in the modelling” – I don’t think you are excluding data points? Rather you are excluding predictions???? In either case, 10% most extreme in what sense/direction???

168: so why include the 10% TP at all?

169: ‘survey’? I don’t think you are actually surveying anything?

172: again the use of ‘file’ is confusing. I think you mean you calculate the vector? The fact that you store this information in a file is incidental surely.

176: “finally, we evaluate the potential migration from one period to the next” – too vague. How did you actually do this based on the centroids explained above??

181: “We focus on the limiting factors affecting the contraction and expansion of climate suitability.” – again, this is vague. What do you mean by this?

186: No, perfect would be AUC=1 ??

188: well predicted rather than well described I think

189: Do these values mean anything? Are they low or high or good or bad?

217: acronyms not useful – explain in full

Discussion.

222-233The content of this paragraph does not refer to results at all. It reads more like general background. It should be in the Introduction, or come later in the Discussion when discussing caveats on the study. The first paragraph of the Discussion should focus on the most interesting results of your study!!

236-241: This is not Discussion, this is Results! The Results section is for presenting results, like this. The Discussion section is for saying why they are interesting, important, relevant, applicable, expected/unexpected, etc etc

255: what’s TS and AP? Overuse of acronyms, especially for Discussion which may often be read in isolation.

258: “the regional heterogeneity of climate baseline and the different amplitude of the variable’s variation” I don’t understand – should be clarified.

Overall the Discussion has many interesting points, but it is much too long and unfocussed. I think it should be substantially rewritten to much better focus on the most important points of YOUR study. For example, 283-290 has no reference to your results at all. In fact 246-290 has only very occasional and brief reference to your results.

The final paragraph on limitations could be much better referenced. The limitations of MaxEnt-style correlative SDMs have been well discussed by many others (eg Fordham below, but many others as well). People are starting to address some limitation using more mechanistic modelling approaches and these should be acknowledged. The CSIRO CLIMEX model is one example, as are the papers below, and there are many others. Some of this work should be acknowledged.


Fordham, D.A., Resit Akçakaya, H., Araújo, M.B., Elith, J., Keith, D.A., Pearson, R., Auld, T.D., Mellin, C., Morgan, J.W., Regan, T.J., Tozer, M., Watts, M.J., White, M., Wintle, B.A., Yates, C., Brook, B.W., 2012. Plant extinction risk under climate change: are forecast range shifts alone a good indicator of species vulnerability to global warming? Glob Change Biol 18, 1357–1371. https://doi.org/10.1111/j.1365-2486.2011.02614.x


Renton, M., Shackelford, N., Standish, R.J., 2012. Habitat restoration will help some functional plant types persist under climate change in fragmented landscapes. Global Change Biology 18, 2057–2070. https://doi.org/10.1111/j.1365-2486.2012.02677.x

Tomlinson, S., Webber, B.L., Bradshaw, S.D., Dixon, K.W., Renton, M., n.d. Incorporating biophysical ecology into high-resolution restoration targets: insect pollinator habitat suitability models. Restor Ecol n/a-n/a. https://doi.org/10.1111/rec.12561

Reviewer 3 ·

Basic reporting

This paper presents a modelling-based study of the historic, present and future distribution of Pseudolarix amabilis, a conifer with a restricted distribution in eastern China. My personal field of expertise is ecology, not modelling, and thus I have appraised the paper on the bases of its ecological relevance and interpretation rather than the technical strength of its modelling approach.

Firstly, the flow and content of manuscript could be improved to help the reader along the journey of the study. At present parts are not concise, particularly the introduction – at four pages long it is a rather rambling background to the study, and several sections could perhaps be removed as they are not intrinsically relevant (LL31-46) or should be consigned to the methods (L101-112). Similarly, the discussion at 11 pages is far too long and reads somewhat like an unfocussed essay. The first paragraph of the discussion makes no reference to the results of the study, which again makes interpretation difficult. Additionally, some elements of the discussion (e.g. L236-241) belong in the results. I was not left with a clear understanding of the study outcomes, how these related to other international literature, or whether the study had achieved its aims of informing future conservation. In fact, the last paragraph seems to suggest that the available data are not sufficient to achieve this aim and that future studies using ENMs might be sufficient. As the reader I need to be led through the study of the study and left with no uncertainty about what was undertaken and why, and the significance of the findings need to be clearly articulated. I am afraid this was not the case – from the summary paragraph (L408-414) I took the message that P. amabilis responds to climatic change exactly as one would expect, and I am afraid I disagree with the statement that “investigating the response mechanism of P. amabilis to past climate change and predicting future climate suitability results in an effective conservation strategy”. This study doesn’t examine the response mechanism of P. amabilis (rather it examines a modelled response underlain by many, many assumptions), and effective conservation requires implementation of actions rather than simply predicting a future scenario.

Fig S1 – very difficult to distinguish between symbols for historical distribution given colour and overlap.

Table S1 – Bio variables should be arranged numerically to match the caption

I think overall, the paper fails this section in its current form.

Experimental design

The paper presents something of a case for conservation, although I feel this basis for the study is rather weak – particularly as the authors don’t really background how threatened the species actually is. The IUCN Red List assessment for P. amabilis notes “this species is very rare in the wild and occurs in a few remnants of primary forest on isolated mountains. Most locations are not within protected areas and loss of habitat is still continuing in this densely populated part of China.” Area of occurrence is estimated at <500 km2, and threatening processes continue unabated. With this in mind, it seems the species requires more pressing conservation initiatives than future modelling – predictions would suggest it may be extinct in the wild long before the 2080 scenario modelled.

Following on from this point, why was the “future moment (2080, HadCM3 A1B)” chosen? There is no justification provided in the text. Is this widely considered to represent the future climate of the region? Does only one modelled snapshot of the future provide an adequate assessment of future conditions and the variability in future climate predictions?

My main issue with the paper is this: the authors provide little discussion of the ecology of the species, which is in my opinion crucial information for the reader in interpreting their results. For example, in the introduction the only non-distributional information about the species (which is only introduced in line 86) is that it belongs to a monotypic genus. We only find out that P. amabilis is a tree as a side note on line 245 in the discussion, and nowhere is it stated that the species is a long-lived conifer restricted to acidic soils between 180-1000 metres altitude (all information provided in the IUCN Red List assessment that is critical to interpreting species distribution). How long-lived is the species? Many conifers are long-lived and take many years to reach reproductive maturity; is 80 years sufficiently far in the future to account for multiple reproductively successful generations? 80 years seems like a very near snap-shot of the future compared with the past in a physiologically adaptive sense (particularly for long-lived individuals): the species had ca. 120,000 years to adapt to changing climate between the LIG and LGM, 21,000 years to adapt between the LGM and present, but only 80 years between the present and the future scenario. I imagine if they are not logged, many individuals alive today will still be alive in 80 years which would imply no change in distribution from climatic factors; rather perhaps a change in the suitability of climate for seedling germination to occur if we work off the assumption that seeds possess a narrow thermal/hydrothermal envelope and exhibit thermal suppression at temperaures 1-2 degrees greater than the present (unlikely based on the global literature). Additionally, there is also no information provided on the dispersal strategy of the species (though there is significant conjecture about past distribution, dispersal, and habitat occupancy).

The authors acknowledge that they do not include soils data in their analyses due to a lack of available data (and note that this omission may limit the efficacy of the model). This is unfortunate given the species has known soil affinities, but is perhaps understandable if such current or historical data is not available. However, the authors also make no mention of altitude or elevation in their manuscript. This to me seems like a terminal oversight given the literature notes the species to be elevational restricted. If the chosen future climate scenario is one in which conditions are warmer than present (this is not clear), then surely an analysis of geographic area incorporating elevational data is required to assess the altitudinal shift in distribution – and this would surely make the situation of the species far more dire.

Validity of the findings

See many of my comments in section 2.

Additional comments

No general comments.

---

## Round 0.2 · accepted · Accept

Thank you for considering PeerJ. You paper has been accepted. Please, consult the uploaded edited file containing my last small edits and submit the final version (following the PeerJ) instructions.

Reviewer 3 ·

Basic reporting

The authors appear to have exhaustively addressed the comments of the reviewers in their response document, and provided significant background to these responses in their rebuttal. The quality of English throughout the manuscript has been improved somewhat. The introduction is to me still rather long, but this is not a huge concern.

The aims and hypotheses are much clearer, and the significance and outcomes of the study are presented with much greater clarity. The addition of ecological information on the species is appreciated and adds to the manuscript, although I feel “branchlets are dimorphic: long branchlets (leading shoots) with helically borne leaves and short branchlets (brachioblasts) with fascicularly arranged leaves.” (L84-86) Is probably unnecessary as morphology is not relevant to the study.

Experimental design

The methods appear well-structured and are presented with a high level of detail. With the additions made after the initial review, I feel these are now satisfactory.

Validity of the findings

I still feel to some extent that this study lacks ecological significance; that is, the data presented are heavily constrained by the absence of soil and altitudinal data given the relevance of these factors on the species distribution. As far as I can see, there is also little attention paid to the highly fragmented nature of remaining natural habitat (very little of which remains in the study region). The authors have acknowledged some of this, and explained why they are not included in the rebuttal, and while I accept this explanation I do wonder how meaningful the conclusions that have been drawn are in their absence. The identification of (for example) 100 km2 of climatically suitable habitat is not particularly useful if half of this is unsuitable soil type, three quarters of the suitable soil type is developed land, and only a fraction of the remaining area is altitudinally suitable. All of these factors are critical for seed germination and seedling establishment, which the authors have noted is the primary life cycle stage they are modelling. As a side note, proper examination and modelling of these factors would require an understanding of seed dormancy, seed ecology, and the thermal niche required for germination to occur, as well as looking at shifts in seasonality which may impact upon moisture availability during critical life cycle stages.

My concern, essentially, is that the authors may be significantly overestimating the future distributional envelope of this species. However, I understand their argument that some of this critical information is not available. I have deliberated on this for some time and cannot decide how to appraise this; I will defer to the opinion of the editor on whether this issue is significant enough to hinder publication.